# Seed Treatments with Microorganisms Can Have a Biostimulant Effect by Influencing Germination and Seedling Growth of Crops

**DOI:** 10.3390/plants11030259

**Published:** 2022-01-19

**Authors:** Mariateresa Cardarelli, Sheridan L. Woo, Youssef Rouphael, Giuseppe Colla

**Affiliations:** 1Department of Agriculture and Forest Sciences (DAFNE), University of Tuscia, 01100 Viterbo, Italy; giucolla@unitus.it; 2BAT Center-Interuniversity Center for Studies on Bioinspired Agro-Environmental Technology, University of Naples Federico II, 80055 Portici, Italy; woo@unina.it; 3Department of Pharmacy, University of Naples Federico II, 80131 Napoli, Italy; 4Task Force on Microbiome Studies, University of Naples Federico II, 80055 Portici, Italy; 5Department of Agricultural Sciences, University of Naples Federico II, 80055 Portici, Italy; youssef.rouphael@unina.it

**Keywords:** seed inoculation, rooting, growth index, mycorrhizae, microbial consortium, *Azotobacter* spp., *Bacillus* spp., *Pseudomonas* spp., cyanobacteria, *Azospirillum* spp., abiotic stress

## Abstract

Seed quality is an important aspect of the modern cultivation strategies since uniform germination and high seedling vigor contribute to successful establishment and crop performance. To enhance germination, beneficial microbes belonging to arbuscular mycorrhizal fungi, *Trichoderma* spp., rhizobia and other bacteria can be applied to seeds before sowing via coating or priming treatments. Their presence establishes early relationships with plants, leading to biostimulant effects such as plant-growth enhancement, increased nutrient uptake, and improved plant resilience to abiotic stress. This review aims to highlight the most significant results obtained for wheat, maize, rice, soybean, canola, sunflower, tomato, and other horticultural species. Beneficial microorganism treatments increased plant germination, seedling vigor, and biomass, as well as overcoming seed-related limitations (such as abiotic stress), both during and after emergence. The results are generally positive, but variable, so more scientific information needs to be acquired for different crops and cultivation techniques, with considerations to different beneficial microbes (species and strains) and under variable climate conditions to understand the effects of seed treatments.

## 1. Introduction

The global agriculture system occupies approximately 40% of the earth’s surface and has the main objective of producing food for an ever-increasing population (7.5 billion people today, and nearly 10 billion by 2050) [1]. Due to the massive use of resources (water, soil, energy, and air), the agricultural sector leaves a large environmental footprint [2] and for this reason, modern technologies and innovative cultivation strategies are almost exclusively aimed at reducing the environmental impact and preserving natural resources for future generations. The main challenges are therefore: (i) to promote the whole soil–plant system equilibrium in space and time, (ii) to reduce nutrient losses from the agroecosystem and to increase soil C sink-potential, (iii) to limit greenhouse gas emissions in order to deal with climate change and to improve system energy use efficiency, and (iv) to ensure high quality production and economic advantages for agricultural producers through an effective and proper use of agronomic inputs [3]. 

For maize, soybean, cereals, rice, cotton, and most vegetable crops the cultivation cycle begins with direct seed sowing; consequently, uniform germination and high seedling vigor contribute decisively to successful crop establishment and thus to crop performance [4]. Indeed, when seedlings emerge fast and vigorously, they have a greater possibility to capture resources, tolerate biotic and abiotic stress, compete with weeds, and, in general, cope better with adverse environmental conditions [5]. Seed quality is therefore a primary objective of the agricultural industry, which reports a significant increase in the global seed market (from USD 30 billion in 1996 to USD 92.32 billion by 2025, according to a new report by Grand View Research, Inc.). To enhance seed quality different seed treatment technologies are now available; in particular, pre-sowing seed treatments arouse great interest because of their efficacy and environmental benefits, plus the field is experiencing large investments by major seed-research and market players. According to market research reports, the global biological seed treatment market is projected to reach USD 1.7 billion by 2025, recording a CAGR of 11.9% since 2020, including biofertilizers, biopesticides, and biostimulants [6]. Considering only seed treatments with microorganisms and natural substances applied as biostimulants, the global market is forecasted to reach USD 338 million by 2025.

Pre-sowing seed treatments with beneficial microorganisms have relatively low application costs, as they require a single treatment and the active ingredients are applied at low dose rates [7,8,9]. These treatments include seed coating and seed priming. Seed coating consists in applying a thin layer of external material onto the seed surface, altering little the seed shape, size, or weight [10,11]. It is generally used for the application of identification colors and tracers (e.g., fluorescent dyes) or active compounds (protectants, plant growth regulating agents, plant nutrients, and microbials) [12]. Some authors prefer to differentiate between ‘seed treatment’ and ‘seed coating’, in order to discriminate between formulations containing useful microorganisms and natural substances in the former and treatments with artificial ingredients such as pesticides in latter [12,13]. Seed treatments do not induce any changes within the seed and the active ingredients provide an advantage for the crop during germination and seedling growth. 

Instead, seed priming is a process of controlled seed hydration with water (hydro-priming) or special solutions (osmo-priming, physicochemical-priming, hormonal priming, or matric-priming) that allows controlled seed imbibition and triggers pre-germinative metabolic processes (de novo synthesis of nucleic acids and proteins, ATP production, sterols and phospholipids accumulation, or activation of DNA repair and antioxidant mechanisms) [14]. Imbibition must be stopped before overcoming the reversible phase, thus prior to emergence of radical from the seed coat [15]. 

Considering the seed industry’s interest on innovative technologies that can make a substantial contribution to the yield and economics, plus the environmental sustainability of the agricultural system, this review aims to focus on the biostimulating effect of beneficial microorganisms applied to the seed of the major crops whose cultivation cycle begins with seed sowing. This effect results in increased plant germination, seedling vigor, and biomass, as well as the capability to overcome seed-related issues (such as abiotic stress) both during and after emergence. 

## 2. Applications of Beneficial Microorganisms

Plant beneficial microbes (PBMs) (growth-promoting rhizo-microorganisms) are specific fungi or bacteria able to establish “intimate” relationships with plants, leading to plant-growth enhancement, increased nutrient uptake, restoration of soil fertility, and improvement of plant resilience to abiotic and biotic stress [16,17,18,19]. Their application to the soil leads to savings in the use of fertilizers and pesticides with considerable agro-ecological advantages [20]. The possibility of applying PBMs directly to the seed produces two substantial benefits, viz. less microbial inoculum per plant located at the seed–soil interface, and immediate contact between microbes and roots at the time of germination and early developmental stages [21,22]. Based on recent research, inoculations refer to seed treatments or biopriming treatments that consist of soaking seeds in a microbial suspension. The microorganisms mainly used for seed treatments belong to arbuscular mycorrhizal (AM) fungi, *Trichoderma* spp., rhizobia, and plant-growth-promoting bacteria (PGPB) (such as *Pseudomonas* spp., *Bacillus* spp., and *Enterobacter* spp.) [14], that have been applied to cereals, vegetables, oil and seed pulse crops, and fiber and forage crops [12]. 

Due to the variability of the experimental conditions, such as the PBMs species-strains, plant species-varieties, cultivation techniques, growing conditions (open field or greenhouse), and climate, the following results have been organized in reference to the most economically important crops for which seeds have been treated using beneficial microorganisms prior sowing.

### 2.1. Wheat

*Triticum aestivum* L. and *T. durum* Desf. are the most widely cultivated wheat species in the world being adapted to a wide range of soil and climate conditions. The cultivation technique for these species has long been characterized by extensive use of agrochemicals and management with a high environmental impact, compromising soil fertility and crop development. On this basis, *T. durum* seeds were treated with a consortium of endophytic microorganisms consisting of *Rhizoglomus intraradices* BEG72 (former *Glomus intraradices*), *Funneliformis mosseae* (former *G. mosseae*), and *Trichoderma atroviride* MUCL 45632 to verify the ability of the fungi to promote emergence and plant growth of seedlings [23] (Table 1). Seventeen days after sowing, a significant effect of the microorganism inoculum was observed in seedlings, with an increase in leaf number (+28.6%), and shoot (+23.1%) and root (+64.2%) dry biomass compared to untreated wheat seedlings. However, there is scientific evidence that different results can be obtained depending on the strain used, as in the case of the experiment carried out by Kthiri et al. [24] on durum wheat (cv. Karim). The authors applied different *Trichoderma harzianum* strains to the outer surface of seeds and obtained different values for germination, seedling growth, and antioxidative system data (phenols and peroxidase enzymes). The strain S.INAT increased root and shoot length, vigor index, and leaf phenolic accumulation with respect to untreated seeds, while the strain S.IO2 induced a higher dry-matter content and peroxidase activity in seedling leaves. Since phenolic compounds and peroxidase enzymes are involved in wall-building processes such as lignification and reinforcement of plant structural components [25], their stimulation via seed treatments can contribute to promote vigor and growth of seedlings [24]. Durum-wheat seeds cv. Karim were also inoculated with *Meyerozyma guilliermondii* yeast, strain S.INAT (MT731365), and sown in pots under controlled conditions. The effect of *M. guilliermondii* yeast on germination and seedling growth was thus verified for the first time. Yeast seed-treatment promoted germination, which increased from 47% (untreated seeds) to 93%, and such growth parameters as shoot and root length and plant biomass. *M. guilliermondii* was able to induce IAA (indole-3-acetic acid) production so promoting plant-cell enlargement, root initiation and lateral-root formation in seedlings [26]. 

External treatments on *T. aestivum* seeds were compared to soil inoculation and foliar spraying in post-emergence by applying mycorrhizal fungus *R. irregularis* and diazotrophic N-fixing bacterium *Azotobacter*
*vinelandii* [27]. The experiment was conducted under controlled conditions and direct seed inoculation led to high root colonization and improved root development through increased root-tip density and branching with respect to other treatments. There is a relationship between root growth and root exudation [28,29] and it is also known that microorganisms modulate root excretion of metabolites in the rhizosphere [30]. Some authors investigated the modulation of root exudation of *T. aestivum* plantlets with respect to the inoculation technique of *T. atroviride* AT10 i.e., with applications of the beneficial fungus in the substrate or as seed treatments [30]. Although no differences were found with respect to inoculation technique for shoot and root dry biomass, the analysis with liquid and gas chromatography coupled to mass spectrometry revealed a distinctive modulation of metabolites (such as lipids, phenols and terpenoids, siderophores and chelating acids, derivatives of amino acids, and phytohormones) in the root exudates depending on the inoculation method. 

To support germination and seedling growth under adverse environmental stress conditions, Lastochkina et al. [31] studied the protective effects of *Bacillus subtilis* (strain 10-4) against drought stress. The bacterium was applied through a bio-priming treatment on seeds of wheat that were sensitive (*T. aestivum* cv. Salavat Yulaev) or tolerant (cv. Ekada 70) to drought conditions during the germination phase. *B. subtilis* promoted germination and plant growth of 6-day-old seedling (both length and fresh/dry weight of roots and shoots) under normal growth conditions and promptly activated specific metabolic adaptations to drought-stress conditions by decreasing lipid peroxidation, proline content, and electrolyte leakage in 21-day-old seedlings [31]. Even *Proteus mirabilis* R2 (LS975374), *Pseudomonas balearica* RF-2 (LS975373), and *Cronobacter sakazakii* RF-4 (LS975370) improved germination of wheat seeds under normal as well as water-stress conditions. Germination and promptness indices reached their highest values (100% and 68%, respectively) for seed treatment with *Pseudomonas*
*balearica,* and even seedling biomass and leaf area were positively affected by the same strain [32]. Otherwise, *Proteus mirabilis* resulted in the highest values of dry weight and leaf area under drought conditions (50% water-holding capacity), which can be attributed to the IAA-production ability of R2 promoting root development and thus nutrient uptake [33,34], and *Cronobacter sakazakii-*treated seeds developed seedlings with minimum cell injury and electrolyte leakage, and maximum cell membrane stability in the presence of drought stress [32]. All bacterial strains induced ACC deaminase and catalase enzyme activity. These are both involved in regulating physiological response to stress since ACC-deaminase regulates the ethylene levels during stress [35] while catalase helps in maintaining reactive oxygen species (ROS) [36,37].

**Table 1 plants-11-00259-t001:** Effect of seed treatments with microorganisms on germination and seedling growth of wheat under different growing conditions.

Crop	Active Microorganisms	Mode of Seed Inoculation; Growth Conditions	Abiotic Stress and References	Main Parameters Improved
Wheat *Triticum durum* Desf	Fungi: *Rhizoglomus intraradices + Funneliformis mosseae + Trichoderma atroviride*	Seed coating; growth chamber	No stress [23]	Increased leaf number (+28.6%), and shoot (+23.1%) and root (+64.25%) dry biomass
	Fungi: *Trichoderma harzianum* (different strains)	Seed coating; growth chamber	No stress [24]	Increased germination (+35%), root (+63%) and shoot (+38%) length, plant dry matter (+550%), vigor index (+120%), and leaf phenols (+128%), according to the strain considered
	Yeast: *Meyerozyma guilliermondii*	Seed coating; growth chamber	No stress [26]	Increased germination (+97%), shoot (+41%) and root (+69%) length, and plant biomass (+16%)
Wheat *T. aestivum* L.	Fungus + bacterium: *Rhizophagus irregularis* + *Azotobacter vinelandii*	Seed coating; rhizoboxes	No stress [27]	Increased root tip density (+28%) and branching (+29%)
	Fungus: *Trichoderma atroviride*	Seed coating; greenhouse	No stress [30]	Different modulation of metabolites (lipids, phenols and terpenoids, siderophores and chelating acids, derivatives of amino acids, and phytohormones) in the root exudates
	Bacterium: *Bacillus subtilis*	Seed priming; growth chamber	Drought [31]	Increased plant elongation (+15%) and plant dry weight (+10%)

### 2.2. Maize

*Zea mays* (L.) is a very important crop for the human diet and as an animal feed [38], and it is grown under a wide range of soil and climatic conditions. Recent research has focused on applying beneficial microorganisms to the seed so that it has a more uniform germination and a better seedling growth. An indigenous strain of *T. harzianum*, for example, proved to be effective as a coating agent of maize seeds in an experiment conducted in open-field conditions (Embu District, Kenya) [39]; whereby inoculated seeds showed higher germination values and 14 days after emergence seedlings had better shoot and root development under *T. harzianum-*coating treatment. Sharma et al. [40] set up an experiment to compare a cyanobacteria consortium (BF1-4) with a biofilm obtained using cyanobacterium *Anabaena torulosa* as matrix and *Trichoderma viride* as partner (An-Tr biofilm), both were applied through priming treatments on two elite maize inbreds (HKI323PV and HKI161PV) (Table 2). The cyanobacterial consortium was comprised of BF1 *A. torulosa*, BF2 *Nostoc carneum*, BF3 *Nostoc piscinale*, and BF4 *Anabaena doliolum*. Germination and germination-related enzymes (α-amylase and invertase activity) were detected 96 h after sowing, highlighting the different effect of BF1-4 and An-Tr biofilm on seeds. Seed inoculation with microorganisms (BF1-4 or An-Tr biofilm) increased germination percentage and enzyme activities for both maize inbreds. The highest germination values were found for inbred HKI323PV seeds inoculated with An-Tr biofilm (16% increase compared to uninoculated control) while for the enzymes involved in the hydrolysis of seed reserve sugars, the greatest increases compared to uninoculated treatment (10 for amylase and 13% for invertase) were found with An-Tr biofilm-based treatment for maize inbred HKI161PV. This improved mobilization of nutrients at the seed stage provides energy for growing of seedlings, as checked at 7 and 21 days after sowing, with significant increases in root length, shoot length, fresh weight, and dry weight upon inoculation with An-Tr biofilm [40].

By using an endophytic bacterial inoculation, higher seed germination and seedling vigor have been observed on young maize plantlets grown in open-field conditions after seed treatment with *Pseudomonas putida, Pseudomonas fluorescens*, *Azospirillum lipoferum,* and *Azospirillum brasilense* [41]. Noumavo et al. [42] considered different seed treatments with *P. putida*, *P. fluorescens,* and *A. lipoferum*, alone or in combination, and the best results for germination percentage, root and shoot length, vigor index, and leaf area were obtained with the combination *P. putida* and *P. fluorescens*. *A. lipoferum* applied alone, on the other hand, stimulated seedling height and root dry matter more than the other treatments [42]. Even under salt-stress conditions, *Pseudomonas* proved to be effective in supporting plant growth and bioaccumulation of natural antioxidant enzymes [43]. The strain (*Pseudomonas genicultate* MF-84) was isolated from maize rhizosphere, tagged with green fluorescent protein for localization in the plant system and used to prime maize seeds; based on observations made with confocal microphotographs, in 15 days *P. genicultate* colonized more than 70% of roots by locating in the epidermal cells, cortical tissues, endodermis, and vascular bundles. 

A seed coating treatment has been performed by using a commercial *A. lipoferum* strain originally isolated from the maize rhizosphere and the sowing experiment was conducted under controlled laboratory conditions. Germination was not affected by *A. lipoferum* treatment but morphological characteristics of the six-leaf-stage seedlings, with longer radicles and larger shoots, revealed the positive impact of *Azospirillum* on early development of plants and, consequently, on leaf photosynthetic potential [44]. Similarly, different isolates belonging to *Bacillus* spp. and characterized for indole-3-acetic acid (IAA) production were applied on maize seeds without leading to differences in germination rate compared to the control but with a significant increase in epigean growth (fresh and dry biomass and shoot length) and root development (fresh and dry biomass and number of adventitious roots) with different results depending on the bacterium strain considered [45]. Accinelli et al. [46] applied a *Bacillus subtilis* strain (QST 713) to maize seeds through the incorporation of spores into a bioplastic-based formulation and only in the presence of the bacterium (but not for treatment with bioplastic film alone) was there a greater elongation of maize seedlings (stems and roots). The application of beneficial microorganisms to the seed via biofilms was also compared with simple seed-coating with adherent spores, using, in this case, two species of *T. harzianum*, and results showed that both *Trichoderma* species promoted seedling growth, especially when applied in a bioplastic layer [47]. For Accinelli et al. [46,47], as for Rozier et al. [44] and Lwin et al. [45], the microorganism treatment did not affect germination percentage but rather, seedlings development and biomass accumulation. On the contrary maize-seed priming by cyanobacterium *Spirulina platensis* accelerated germination in both control and Cd-toxicity conditions with a maximum in the absence of Cd contamination. Even shoot length and photosynthetic capacity of seedlings (measured 20 days after sowing) were enhanced by the *S. platensis* treatment whereas Cd accumulation and its translocation from root to shoot was significantly restricted after maize-seed inoculation with cyanobacterium, pointing out the value of seed treatment with *S. platensis* in cropping systems conducted on contaminated soil [48]. Recently, an experiment was conducted with the aim of testing the effect of the endophytic bacterium *Mixta theicola* isolated from roots of *Solenostemma argel*, a wild herb, in maize seed priming. Inoculation resulted in a significant enhancement of seed germination, root elongation, seedling vigor index, and fresh and dry biomass of 15-day-old plantlets. Even chlorophyll, carbohydrates, proteins, anthocyanins, total phenolics, and flavonoids were increased in the inoculated seedlings indicating an improved physiological and biochemical status of maize seedlings [49]. 

**Table 2 plants-11-00259-t002:** Effect of seed treatments with microorganisms on germination and seedling growth of maize (*Zea mays* L.) under different growing conditions.

Active Microorganisms	Mode of Seed Inoculation; Growth Conditions	Abiotic Stress and References	Main Parameters Improved
Cyanobacteria and fungus: *Anabaena torulosa + Nostoc carneum + Nostoc piscinale + Anabaena doliolum* or *Anabaena torulosa* + *Trichoderma viride*	Seed coating; greenhouse	Arsenic [37]	Germination (+16%) and seed-germination-related enzymes (+10% for α-amylase and + 13% for invertase), root length (+43%), shoot length (+90%), fresh weight (+21%), and dry weight (+31%)
Bacteria: *Pseudomonas putida*, *Pseudomonas fluorescens, Azospirillum lipoferum*, alone or in combination	Seed coating; growth chamber	No stress [42]	Germination (+22%), root length (+51%), shoot length (+54%), vigor index (+75%), and leaf area (+86%)
Cyanobacterium: *Azospirillum lipoferum*	Seed coating; growth chamber	Nitrogen [44]	Radicle (+36%) and shoot biomass (+30%)
Bacterium: *Bacillus* spp.	Seed soaking; Growth chamber	No stress [45]	Shoot fresh (+90%) and dry (+91%) biomass, shoot length (+37%), root fresh (+88%) and dry (+69%) biomass, and number of adventitious roots (+61%)
Bacterium: *Bacillus subtilis*	Seed coating (bioplastic formulation); growth chamber	No stress [46]	Shoot (+7%) and root (+10%) length
Fungus: *Trichoderma harzianum*	Seed coating (and biofilm application); growth chamber	No stress [47]	Shoot (+14%) and root (+9%) length
Bacterium: *Mixta theicola*	Seed soaking; growth chamber	No stress [49]	Germination (+38%), root elongation (109%), seedling-vigor index (+117%), and fresh (+108%) and dry (+207%) biomass
Cyanobacterium: *Spirulina platensis*	Seed priming; growth chamber	Cadmium [48]	Germination (+63%), root dry weight (+57%), and leaf area (+20%)

### 2.3. Rice

Rice (*Oryza sativa* L.) is one of the most widely consumed grains in the world, with high water requirements and a distinctive cultivation technique. At present, dry-direct seeding is preferred as it reduces labor and water consumption, but it means more weed competition during the first developmental stage, which can reduce the crop yield by 30–80%. To promote seedling emergence and development after dry-direct seeding, Javed et al. [50] applied plant-growth-promoting bacteria (*Bacillus* sp. KS-54) to the seeds with coating technology (Table 3). Seed coating with *Bacillus* sp. KS-54 was effective in both controlled and open-field conditions compared to noncoated seeds—in the first case the coating treatment enhanced final germination and lowered mean germination time, whereas in field conditions it increased emergence and values of the emergence index, representing a fast and synchronized germination [50]. Even the simultaneous application of *Pseudomonas* and *Bacillus* increased the rice seed vigor index, as reported by Palupi et al. [51]. A coating treatment with *Paenibacillus yonginensis* bacterium combined with SiO_2_ resulted in an increase in all seedling growth parameters (shoot length, root length, root number, fresh weight, and dry weight) and vigor index estimated by means of seedling length and germination [52]. 

### 2.4. Soybean

*Glycine max* ((L.) Merr) is an important crop worldwide. Its development and yield in open fields are strongly influenced by the symbiosis with soil microbes. However, there are not many trials involving seed treatments with microorganisms and natural substances, although soybean plants are naturally found associated with arbuscular mycorrhizal fungi and rhizobia [53]. Yusnawan et al. [54] analyzed how seed coating with different *Trichoderma virens* isolates affects germination and development of soybean seedlings (Table 4). Germination was inhibited by five of the seven strains tested and no strain improved the final germination percentage compared to uncoated seeds. Some *T. virens* isolates positively affected shoot and root length whereas all *T. virens* strains increased root weight. Total flavonoid and phenolic content of 14-day-old seedlings was also affected by seed treatments revealing a significant stimulation of secondary metabolism in the presence of *Trichoderma* inoculum. *Bacillus velezensis* strain CMRP 4490 was applied as a coating film on soybean seeds and the germination rate increased compared to the control from 55.5 to 64% [55]. The CMRP 4490 treatments resulted in differences in total root length and total root surface and strain genome exploration revealed the presence of genes linked to the regulation of biofilms, motility, and important properties for rhizospheric colonization and plant-growth-promoting ability [55]. Jarecki [56] demonstrated the effectiveness of *Bradyrhizobium japonicum* as a seed inoculant for increasing root nodulation of soybean. The same bacterium, either alone or in combination with a mycorrhizal inoculum (*Glomus clarum*, *Glomus mosseae,* and *Gigaspora margarita*), improved root length and plant dry-weight as compared to untreated plants [57]. 

### 2.5. Canola

Canola (*Brassica napus* L.) has become one of the world’s most important oilseed crops. With the aim to promote canola-seedling growth in open-field conditions, Noel et al. [58] evaluated the effectiveness of seed inoculation with *Rhizobium leguminosarum* as a plant-growth-promoting rhizobacterium and achieved early stimulation of seedling root growth (Table 4). Direct involvement of the plant growth regulators indole-3-acetic acid and cytokinin was confirmed by using an auxotrophic *Rhizobium* mutant that requires tryptophan or adenosine for indole-3-acetic acid and cytokinin synthesis, respectively, and did not promote seedling roots as would be the case in a non-auxotrophic *Rhizobium* strain [58]. As for maize, Accinelli et al. [46] obtained a significant stimulation of canola seedling growth, but not of germination rate, after seed-coating treatments with *T. harzianum*. Under salt stress, biopriming treatments with *Bacillus subtilis* (bacterium), *Macrophomina phaseolina* (fungus), or a combination of both, enhanced germination parameters (percentage and velocity) of canola seeds, even under high salinity conditions [59]. Similarly, *B. subtilis* and *Trichoderma harzianum* increased germination as well as root length and seedling vigor index if applied as a coating to canola seeds [60].

### 2.6. Sunflower

Sunflower (*Helianthus annuus* L.) seeds have been treated with *Enterobacter* (FD-17), *Bacillus* sp. (KS-54) and *Paraburkholderia phytofirmans* (PsJN) because these microorganisms contribute to nutrient mineralization and solubilization and could improve seed germination [61] (Table 4). According to results, *Enterobacter* and *P. phytofirmans* optimized germination percentage, mean growth time, and vigor indices of the biochemical profile of sunflower seedlings (activity of catalase, peroxidase, superoxide dismutase, protease, α-amylase enzymes; total soluble proteins; and lipid peroxidation). Even thirty *Pseudomonas fluorescens* strains were applied as coating agents improving seed germination and the vigor index of sunflower seedling over the control (untreated seeds) [62]. *Pseudomonas fluorescens* strains were also evaluated for their ability to enhance shoot and root length, lateral rooting, and biomass with positive results for all growth indices measured on sunflower seedlings.

**Table 4 plants-11-00259-t004:** Effect of seed treatments with microorganisms on germination and seedling growth of industrial crops under different growing conditions.

Crop	Active Microorganisms	Mode of Seed Inoculation; Growth Conditions	Abiotic Stress and Reference	Main Parameters Improved
Soybean *Glycine max* (L.) Merr	Fungus: *Trichoderma virens*	Seed coating	No stress [54]	Shoot (+16%) and root (+37%) length, root weight (+77%), and shoot weight (+25%)
	Bacterium: *Bacillus velezensis*	Seed coating; growth chamber	No stress [55]	Germination rate (+15%), total root length (+33%), and total root surface (+27%)
Canola *Brassica napus* L.	Bacterium: *Bacillus subtilis*	Seed coating (bioplastic formulation); growth chamber	No stress [46]	Shoot (+15%) and root (+12%) length
	Bacterium and fungus: *Bacillus subtilis*, *Macrophomina phaseolina*, alone or in combination	Seed priming; growth chamber	Salt [59]	Germination
	Bacterium and fungus: *Bacillus subtilis* + *Trichoderma harzianum*	Seed priming; growth chamber	Salt [60]	Germination
Sunflower *Helianthus annuus* L. L.	Bacteria: *Enterobacter*, *Bacillus* sp., *Paraburkholderia phytofirmans*	Seed priming; growth chamber	No stress [61]	Germination and vigor index
	Bacterium: *Pseudomonas fluorescens*	Seed priming; growth chamber	No stress [62]	Germination and vigor index

### 2.7. Tomato

Tomato is a very important vegetable crop, grown worldwide in outdoor fields, greenhouses, and net houses. Mastouri et al. [63] considered different abiotic stress conditions during tomato seed germination and evaluated the effect of *Trichoderma* seed treatment as means to maintain good seedling performance even in such conditions. *Trichoderma* treatment did not have an effect in absence of stress but under osmotic, salt, or suboptimal temperature conditions it guaranteed a faster and more uniform germination with respect to the control. A common mechanism through which the plant–fungus association enhances tolerance to different abiotic stresses was postulated—*Trichoderma* could induce some physiological protection in stressed seedling through a reduced accumulation of lipid peroxides in the presence of oxidative damage. Seed biopriming with *Trichoderma pseudokoningii* in combination with vermiwash, increased root biomass of tomato seedling under heat-stress conditions [64]. A coating inoculum of *T. harzianum* and *P. fluorescens* (either singly or in combination) induced a significantly higher germination rate (more than 48%) and a lower mean germination time (less than 2.5 days) of tomato seeds; the combinations of inoculants were more effective than single-isolate treatments [65] (Table 5). 

### 2.8. Other Horticultural Species

There are some cases in which seed treatments induced negative effects on germination rate and plant height. This is the case of seed coating by using the consortium of *Trichoderma* spp., *Beauveria bassiana*, *Metarhizium anisopliae*, and arbuscular mycorrhizal fungi that reduced the germination of lettuce seeds [66]. Similar results were obtained by coating sweet pepper seeds with *T. viride*, *T. polysporhum*, *T. stromaticum*, *B. bassiana*, *M. anisopliae*, and arbuscular mycorrhizal fungi [12]. It should be noted that *Trichoderma* produces and secretes a wide range of extracellular hydrolytic enzymes capable of degrading plant cell walls; if applied in in high doses, it can therefore attack the seed tegument, damaging it and causing a reduction in germination and plant growth [67,68]. Carrot and onion seeds primed with beneficial microorganisms (*Clonostachys rosea*, *Pseudomonas chlororaphis*, *Pseudomonas fluorescens*, *T. harzianum,* and *T. viride*) were sown in glasshouse experiments and displayed improved emergence of treated carrot seeds and a better emergence time for *C. rosea* coating treatment [69] (Table 5). Considering cucumber seeds, Pill et al. [70] applied a commercial preparation of *T. harzianum* as a coating agent obtaining higher seedling emergence and seedling shoot fresh weight with respect to uncoated seeds. The same *Trichoderma* species was effective for ameliorating germination and seedling growth of *Cucurbita pepo* under different salinity stresses (50 and 100 mM NaCl solution) [71] and of *Cuminum cyminum* under drought stress [72]. Piri et al. [72] applied different *T. harzianum* strains or *Pseudomonas fluorescence* bacteria and observed an increase in soluble protein and antioxidant enzyme activity of inoculated seeds with respect to a control without inoculation leading to enhancement of some morphological indices. 

**Table 5 plants-11-00259-t005:** Effect of seed treatments with microorganisms on germination and seedling growth of vegetable crops under different growing conditions.

Crop	Active Microorganisms	Mode of Seed Inoculation; Growth Conditions	Abiotic Stress and References	Main Parameters Improved
Tomato *Lycopersicon esculentum* Mill.	Bacterium and fungus: *Trichoderma harzianum*, *Pseudomonas fluorescens.* alone or in combination	Seed coating; growth chamber	No stress [65]	Germination rate (+48%)
	Fungus: *Trichoderma pseudokoningii*	Seed priming; growth chamber	Heat [64]	Shoot (+169%) and root (+135%) length, root number (+77%), shoot (+26%) and root (+54%) fresh weight, and shoot (+131%) and root (+276%) dry weight
Cucumber *Cucumis sativus* L.	Fungus: *Trichoderma harzianum*	Seed coating; growth chamber	No stress [70]	Seedling emergence and shoot fresh weight
Carrot *Daucus carota* L. and onion *Allium cepa* L.	Bacteria and fungi: *Clonostachys rosea*, *Pseudomonas chlororaphis*, *Pseudomonas fluorescens*, *Trichoderma harzianum, Trichoderma viride*	Seed priming; greenhouse	No stress [69]	Emergence

## 3. Conclusions

Since seed quality is a primary objective of the agricultural industry and different seed-treatment technologies are now available, it is interesting to consider the effectiveness of beneficial microorganisms for seed treatments of the main crops whose cultivation cycle begins with direct seed sowing. In fact, both germination and seedling vigor contribute to successful crop performance because rapidly emerging healthy seedlings are able to immediately utilize the available resources, tolerate biotic and abiotic stresses and other adverse environmental conditions. We examined how plant beneficial microbes belonging to arbuscular mycorrhizal fungi, *Trichoderma* spp., rhizobia, and other bacteria improve germination and emergence performance of crops. In addition to agronomic benefits, the possibility of applying beneficial microorganisms directly to the seed allows for reduction of the amount of microbial inoculum per plant (economic advantage) and ensures an early contact between microbes and rootlets. Until now, the most significant results have been obtained for wheat, maize, rice, soybean, canola, sunflower, tomato, and other horticultural species. More research is needed to explore new crops and microorganism strains and to deepen scientific information on different climate and growing conditions. 

Currently, only mycorrhizal fungi, *Rhizobium* spp., *Azotobacter* spp., and *Azospirillum* spp. are included in EU Fertilizer Regulation 2019/1009 (Component Material Categories, number 7) (CMC-7) as microbial plant biostimulants. With regard to other microorganisms in this review, currently *Trichoderma* spp. is registered as a microbial biological control agent, as a biopesticide against plant pathogens. However, this fungus, and numerous other microorganisms are found frequently as active ingredients in many commercial formulations with indications as biofertilizers, bio-growth enhancers, and biostimulants due to the ability of these biological components to promote plant growth, abiotic stress tolerance, improve yield and nutritional quality, as proven in various crop studies. The above findings presented suggest that newly selected microbial organisms should be considered for inclusion in revised versions of the CMC-7 list, according to their ability to establish broad and positive relationships with plants, even under suboptimal or stressful environmental conditions, and to enhance yield and quality. However, the future inclusion of new microbial strains will require sufficient scientific evidence that demonstrates and supports not only their efficacy, but also their safety to the consumers and environment.

## Figures and Tables

**Table 3 plants-11-00259-t003:** Effect of seed treatments with microorganisms on germination and seedling growth of rice (*Oryza sativa* L.) under different growing conditions.

Active Microorganisms	Mode of Seed Inoculation; Growth Conditions	Abiotic Stress and References	Main Parameters Improved
Bacterium: *Bacillus* sp.	Seed coating; growth chamber	Submersion [50]	Germination
Bacterium: *Paenibacillus yonginensis*	Seed soaking; growth chamber	No stress [52]	Germination (+4%), shoot length (+14%), root length (+26%), root number (+46%), and seedling fresh weight (+9%)

## Data Availability

Not available.

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
