# Peer review of "Seed Treatments with Microorganisms Can Have a Biostimulant Effect by Influencing Germination and Seedling Growth of Crops"

_plants, 2022, doi:10.3390/plants11030259_

Round 1

Reviewer 1 Report

This is a timely and interesting summary of previous studies of the effect of seed innocula on germination growth of several crops.  The manuscript is thorough and well-written.  Its main weakness is that it concentrates on describing previous results but does not make significant effort to generalize or synthesize these information.  Nevertheless, it is a thorough summary and will a useful contribution to the field.

One minor comment:  this sentence seems out of place in its present context and should be explained further or places elsewhere:

  1. Since phenols and peroxidases enzymes are involved in lignification and structural reinforcement of plant cell-wall [22], their stimulation via seed treatments can contribute to promote seedling vigor and growth

A few minor typos were detected:

OLD:

133 production so promoting plant cell enlargement, toot initiation and lateral root formation

NEW:

133 production so promoting plant cell enlargement, root initiation and lateral root formation

OLD:

178 conditions (Embu District, Kenya) [36]; inoculate seeds showed higher germination values

NEW:

178 conditions (Embu District, Kenya) [36]; inoculated seeds showed higher germination values

OLD:

197 By using endophytic bacterial inoculation, it has been observed higher seed germi-

NEW:

197 By using endophytic bacterial inoculation, higher seed germination has been observed

OLD:

347 treatment technologies are now available, it’ interesting to consider the effectiveness of

NEW:

347 treatment technologies are now available, it’s interesting to consider the effectiveness of

Author Response

dear Reviewer

we appreciate your comments and according to your suggestions we improved the manuscript, even generalizing some information.

Detailed answers are reported in a point-to-point manner.

‘This sentence seems out of place in its present context and should be explained further or places elsewhere: L124 (Since phenols and peroxidases enzymes are involved in lignification and structural reinforcement of plant cell-wall [22], their stimulation via seed treatments can contribute to promote seedling vigor and growth)’

We better explain the sentence (L124-131)

OLD: 133 production so promoting plant cell enlargement, toot initiation and lateral root formation
NEW: 133 production so promoting plant cell enlargement, root initiation and lateral root formation
OLD: 178 conditions (Embu District, Kenya) [36]; inoculate seeds showed higher germination values
NEW: 178 conditions (Embu District, Kenya) [36]; inoculated seeds showed higher germination values
OLD: 197 By using endophytic bacterial inoculation, it has been observed higher seed germi-
NEW: 197 By using endophytic bacterial inoculation, higher seed germination has been observed
OLD: 347 treatment technologies are now available, it’ interesting to consider the effectiveness of
NEW: 347 treatment technologies are now available, it’s interesting to consider the effectiveness of

Concerning these minor typos, we made the appropriate corrections.

Reviewer 2 Report

The manuscript is written correctly and comprehensively.
I have no comments.

Author Response

dear Reviewer

thank you for your contribution and your time

Reviewer 3 Report

The manuscript describes the results of studies on microbiological seed treatment. This is in line with the authors' declaration (Mini review). The described results include selected manuscripts (references), but the most important ones. Supplementing requires inoculation of soybean seeds with Bradyrhizobium japonicum bacteria. I have included other comments in the text. After corrections, I recommend that the manuscript (Mini review) be published in Plants journal. 

Author Response

dear Reviewer

we appreciate your comments and according to your suggestions we improved the manuscript.

We inserted the results reported for treatments to soybean seeds with Bradyrhizobium japonicum bacteria (L288-292). We are sorry for not considering it from the beginning.

We have also solved all the comments you kindly made in the pdf.

here are some considerations reported in a point-to-point manner:

  • In the review we considered only abiotic stress (answer to comment in L23);
  • About comment in L 53 we modified the text (considering that seed treatments are explained below)
  • Comment L84: solved

Reviewer 4 Report

The manuscript entitled “Seed treatments with microorganisms have a biostimulant effect by influencing germination and seedling development of 3 different crops”, authored by Mariateresa Cardarelli, Sheridan L. Woo, Youssef Rouphael, and Giuseppe Colla, deals with the reviewing of the potential biostimulant effect of the treatments with beneficial microbes. The work is well written, however, the presence of some shortcomings in the manuscript make it mandatory major changes.

First of all, because the limited information included in the reviewed manuscript, this reviewer understands the authors' intentions to classify their own manuscript as “mini review”. However, this type of manuscript is not included among those accepted by the MDPI system. Authors should refer to author guidelines (specifically, to this section: https://www.mdpi.com/about/article_types) and choose a more relevant type of manuscript. I suggest to consider this manuscript as an “Opinion” paper, or at most a “Perspective” paper.

Moreover, I strongly suggest changing the title of the manuscript. Indeed, the actual title of the manuscript mistakenly suggest that all microorganisms applied via seed treatment display biostimulating effects. This affirmation, as claimed by the same authors, is not always true and it is very variable. It mainly depends on the type of inoculated microorganism, seed species, experimental conditions, and kind of stress to which the different seeds are exposed.

Keywords should be words not contained in the title or in the abstract. Their usefulness is to make easier the searching of the article using the common scientific search engines. Since several keywords are already present in the title, and/or repeated several times in the abstract section, I strongly advise the authors to replace some of them and add more. As journal guidelines clearly report, a limited number of keywords can be used (maximum 10). Consequently, authors should carefully choose them.

Please carefully check the references style in the manuscript. Two different formats are currently used. For example: reference in line 60 has a different reference style.

In this review, the authors discuss the potential biostimulating effect of microorganisms applied as seed treatment on four different crops. However, a very important aspect is not discussed. In particular, the authors should specify that exclusively microorganism or a consortium of microorganisms listed in the CMC-7 (Component Material Categories, number 7) can be currently used as biostimulants. This list, which includes only four genera (Azotobacter spp., Mycorrhizal fungi, Rhizobium spp., and Azospirillum spp), is a big limitation in this experimental field. This is the main reason why other microorganisms are not currently used as biostimulants in agriculture, although other strains may show equally positive and beneficial effects. Although having regulations and limitations may be useful for guaranteeing food security and quality, the stringency and exclusivity of the positive list may strongly affect the potential benefits of these new products. Consequently, it may be appropriate to consider the reduction of the negative list and the expansion of the positive one with new microbial organisms, assuming scientific evidence can demonstrate and support their safety for both the environment and consumers (REFERENCE: Microbial Biostimulants as Response to Modern Agriculture Needs: Composition, Role and Application of These Innovative Products; DOI: https://doi.org/10.3390/plants10081533).

When we talk about biostimulants, we mainly refer to the use of formulation that help plants in particular abiotic stress conditions. The authors should insert in the presented tables an additional column, in which the type of abiotic stress of the respective reference is specified.

Author Response

dear Reviewer

we appreciate your comments and according to your suggestions we improved the manuscript.

Detailed answers are reported in a point-to-point manner.

‘Keywords should be words not contained in the title or in the abstract’

We made the appropriate text corrections.

Please carefully check the references style in the manuscript. Two different formats are currently used. For example: reference in line 60 has a different reference style.

We corrected the style. Sorry for the oversight

In this review, the authors discuss the potential biostimulating effect of microorganisms applied as seed treatment on four different crops. However, a very important aspect is not discussed. In particular, the authors should specify that exclusively microorganism or a consortium of microorganisms listed in the CMC-7 (Component Material Categories, number 7) can be currently used as biostimulants. This list, which includes only four genera (Azotobacter spp., Mycorrhizal fungi, Rhizobium spp., and Azospirillum spp), is a big limitation in this experimental field. This is the main reason why other microorganisms are not currently used as biostimulants in agriculture, although other strains may show equally positive and beneficial effects. Although having regulations and limitations may be useful for guaranteeing food security and quality, the stringency and exclusivity of the positive list may strongly affect the potential benefits of these new products. Consequently, it may be appropriate to consider the reduction of the negative list and the expansion of the positive one with new microbial organisms, assuming scientific evidence can demonstrate and support their safety for both the environment and consumers (REFERENCE: Microbial Biostimulants as Response to Modern Agriculture Needs: Composition, Role and Application of These Innovative Products; DOI: https://doi.org/10.3390/plants10081533).

we thank the referee for these insightful considerations on the basis of which we have significantly improved the manuscript by including comments and information (L381-394)

When we talk about biostimulants, we mainly refer to the use of formulation that help plants in particular abiotic stress conditions. The authors should insert in the presented tables an additional column, in which the type of abiotic stress of the respective reference is specified.

We made the appropriate table corrections.

Moreover, I strongly suggest changing the title of the manuscript. Indeed, the actual title of the manuscript mistakenly suggest that all microorganisms applied via seed treatment display biostimulating effects. This affirmation, as claimed by the same authors, is not always true and it is very variable. It mainly depends on the type of inoculated microorganism, seed species, experimental conditions, and kind of stress to which the different seeds are exposed.

We made the appropriate title changes.

dear Reviewer

we kindly ask you to re-evaluate the ‘Review’ option for this manuscript. As requested by the MDPI system, the manuscript offer a comprehensive analysis of the extant literature related to seed treatments with beneficial microorganisms and their efficacy on germination and seedling growth (at least 4000 words).

More frequently, research activities focused on plant development, yield, quality and phytosanitary aspects of crops following treatments (including seed treatments). Not many studies focused on germination response and seedling vigor and development even if seed quality is a primary objective to keeping an uniform germination and high seedling vigor (better crop establishment and crop performance: higher ability to capture resources, tolerate stress, compete with weeds …

Round 2

Reviewer 4 Report

The authors correctly addressed all my concerns. Consequently I think that the manuscript is now suitable as publication.